# A Review of Evapotranspiration Measurement Models, Techniques and Methods for Open and Closed Agricultural Field Applications

**Ikhlas Ghiat, Hamish R. Mackey** and **Tareq Al-Ansari** *

College of Science and Engineering, Hamad Bin Khalifa University, Qatar Foundation, Doha P.O. Box 5825, Qatar; ighiat@hbku.edu.qa (I.G.); hmackey@hbku.edu.qa (H.R.M.)
* Correspondence: talansari@hbku.edu.qa

**Abstract:** Detailed knowledge of energy and mass fluxes between land and the atmosphere are necessary to monitor the climate of the land and effectively exploit it in growing agricultural commodities. One of the important surface land fluxes is evapotranspiration, which combines the process of evaporation from the soil and that of transpiration from plants, describing the movement of water vapour from the land to the atmosphere. Accurately estimating evapotranspiration in agricultural systems is of high importance for efficient use of water resources and precise irrigation scheduling operations that will lead to improved water use efficiency. This paper reviews the major mechanistic and empirical models for estimating evapotranspiration including the Penman–Monteith, Stanghellini, Priestly–Taylor, and Hargreaves and Samani models. Moreover, the major differences between the models and their underlined assumptions are discussed. The application of these models is also reviewed for both open and closed field mediums and limitations of each model are highlighted. The main parameters affecting evapotranspiration rates in greenhouse settings including aerodynamic resistance, stomatal resistance and intercepted radiation are thoroughly discussed for accurate measurement and consideration in evapotranspiration models. Moreover, this review discusses direct evapotranspiration measurements systems such as eddy covariance and gas exchange systems. Other direct measurements appertaining to specific parameters such as leaf area index and surface leaf temperature and indirect measurements such as remote sensing are also presented, which can be integrated into evapotranspiration models for adaptation depending on climate and physiological characteristics of the growing medium. This review offers important directions for the estimation of evapotranspiration rates depending on the agricultural setting and the available climatological and physiological data, in addition to experimentally based adaptation processes for ET models. It also discusses how accurate evapotranspiration measurements can optimise the energy, water and food nexus.

**Keywords:** evapotranspiration; greenhouse; agriculture; energy; water; food; nexus

## 1. Introduction

The global increase in food demands pressures food systems to increase yields despite limitations in water resources. As such, there is an impetus to move to more sustainable practices and optimised operations for agricultural systems that will enable efficient use of water resources [1]. A key aspect for efficient agricultural practices is adequate irrigation management, which depends on accurate estimates of crop water requirements. Evapotranspiration (ET) is a measure of crop water requirements, which entails vapour water movement from the land to the atmosphere in the form of evaporation from the soil and transpiration from the plants [2]. Hence, the appropriate evaluation of evapotranspiration is necessary to prevent excess or deficit irrigation and sustain the use of water resources while offering the necessary agricultural commodities. This can be achieved through models that measure and predict evapotranspiration rates, or direct measurements using

high-performing instruments. Challenged by the complexity and high cost of directly measuring evapotranspiration, numerous efforts have been deployed in developing estimation models that can easily be applied to varied applications and growing mediums [3].

The accurate estimation of crop water requirements is of high importance in the agricultural sector as it aids in the optimal operations of irrigation scheduling in terms of frequency and quantity. Knowledge of evapotranspiration rates can hence support growers to meet their cultivation targets, improve water use efficiency, increase crop yields, reduce energy consumption, and reduce associated environmental emissions [4]. This indicates apparent intertwined interlinkages between the energy water and food (EWF) systems that are driven by ET estimates and measurements. The EWF nexus is a holistic approach that aims to evaluate the inherent interdependencies between the energy, water and food systems, and identify trade-offs and synergies between their resources [5]. Through this, the EWF nexus approach enables the optimisation of resource consumption and the reduction of associated system environmental burdens [6–8]. Thus, it is necessary for agricultural food systems to adopt an EWF nexus methodology to ascertain a sustainable intensification that can meet the growing population demand for nutritious food, conserve water and energy resources, and preserve the environment from further degradation [9].

With regard to evapotranspiration models, the first ET model was developed by Penman, in which only external physical drivers were considered [10]. This latter model was further improved by Monteith, who integrated physiological characteristics [11]. Other simplified versions of the Penman–Monteith equation were proposed, which require less input data [12]. Since these models have been developed under specific meteorological and physiological conditions and for specific settings (i.e., open or closed fields), it is necessary to choose the model with the closest conditions and assumptions as the system under study. Moreover, adapting these models to the specific growing conditions of the evaluated system can enhance the accuracy of ET estimates. This can be achieved through the parametrisation of observed relationships through direct measurements [13]. The vast majority of reviews conducted around evapotranspiration address the main differences between certain empirical ET models in terms of input data, accuracy, and limitations [3,14,15]. Others only reviewed specific measurement techniques such as remote sensing or for assessing certain parameters such as the leaf area index [16,17]. However, there is a lack of studies that discuss the applicability of ET models to open and closed agricultural mediums, and that provide methods and directions for ET estimate improvement through direct measurements. As such, this review aims at responding to this gap by aggregating ET models and measurement methods in one study, and evaluating their applicability for different agricultural settings as well as discussing ET estimates from an EWF nexus perspective.

The objective of this study is to review the major mechanistic and empirical models for the estimation of evapotranspiration and evaluate their applicability for different agricultural systems. The following models are considered: (1) Penman–Monteith; (2) FAO56 Penman–Monteith; (3) Stanghellini; (4) Priestley–Taylor; and (5) Hargreaves and Samani. Moreover, this review aims at presenting different measurement systems for the estimation of important parameters that are required by the models including leaf area index and surface leaf temperature, along with direct ET measurement techniques. As illustrated in Figure 1, the two review sections provide the basis for general directions to evaluate ET rates for both open and closed (greenhouse) agricultural settings. This paper is divided into four sections: section one discusses the importance of accurate ET estimates in the optimisation of the energy, water and food (EWF) nexus, section two reviews mechanistic and empirical ET models, section three discusses ET measurement systems, and section four presents directions for accurate ET estimates depending on the application.

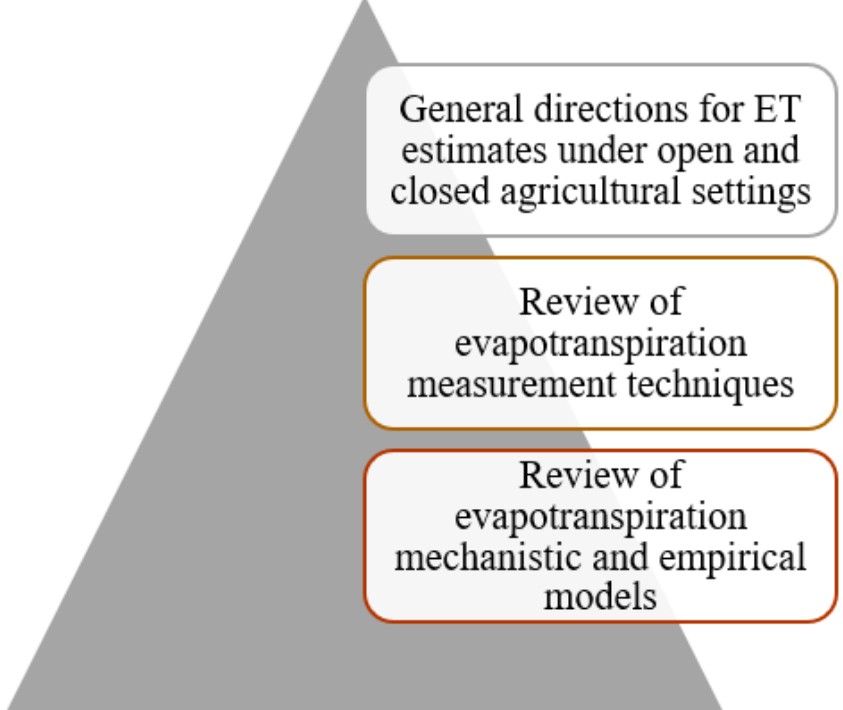

**Figure 1.** Review methodology.

## 2. Research Methods

The presented review examines a series of peer-reviewed journal articles, mostly from the year of 1948 which corresponds to the first developed ET model until 2021, accessed through recognised databases including ScienceDirect, Wiley Online Library, and Springer, etc. Additionally, other peer-reviewed sources are also considered including public reports, conference proceedings and book chapters in efforts to enrich the quality of this review. This work focuses on the major evapotranspiration models and measurement techniques and proposes general directions for ET estimation for open and closed agricultural mediums. In addition, it examines the role of ET estimates in optimising the energy, water and food nexus. Thus, the selected literature is classified into three major categories: studies tackling (1) ET models and their applicability; (2) ET measurement systems; and (3) the role of ET in EWF nexus optimisation. The search method used for the identification of studies is based on searching a diverse range of terms related to the three aforementioned categories in the keywords, title and/or abstract of the articles. Figure 2 provides a network visualisation of the main keywords used in the search and that co-occur in most of the chosen articles. The higher the occurrence of the term in the title and abstract of the articles, the larger the circle and label appear. Four colours are used in this visualisation with each representing a cluster. Clusters encompass terms that have a high correlation with each other. The lines represent co-occurrence links between two terms. As the number of publications where the two terms occur together increases, the link appears stronger.

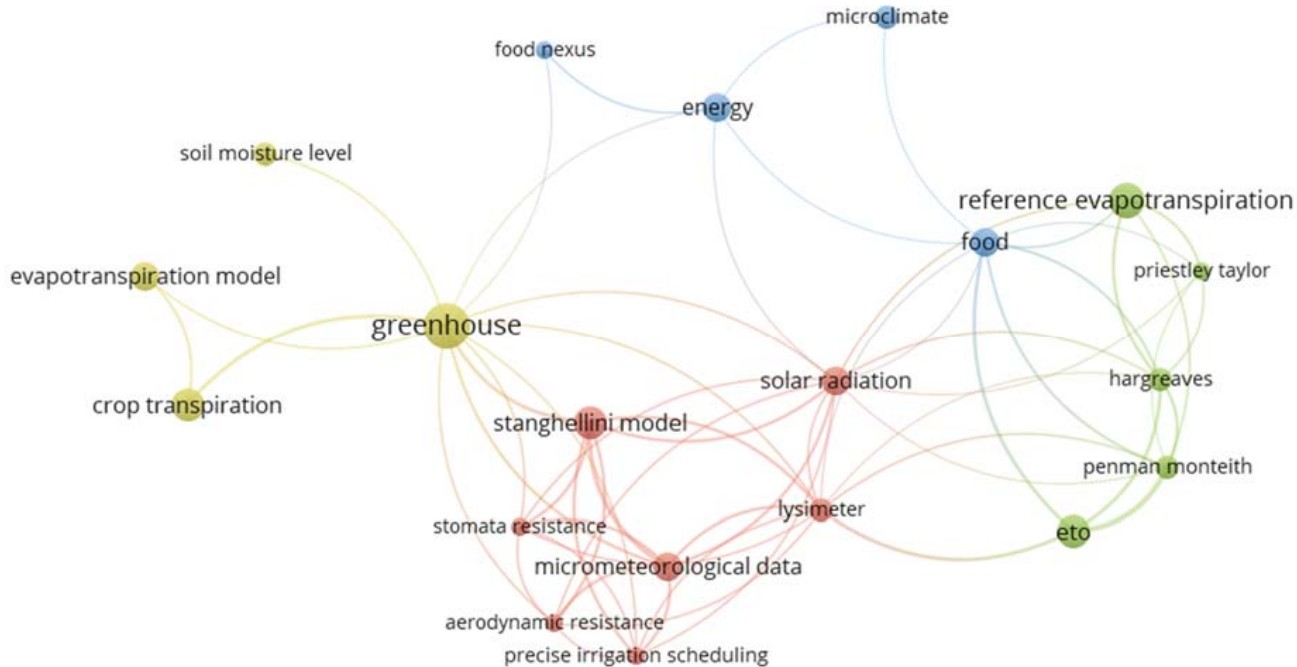

**Figure 2.** Network visualisation of search keywords.

### 3. The Role of Evapotranspiration Measurement in Optimising the Energy, Water and Food (EWF) Nexus

The continuously growing population and the rising economic growth engender many challenges in securing the intensified energy, water and food demands. The intertwined dependence between the energy, water and food sectors makes it imperative to adopt a holistic nexus approach in confronting these challenges [5]. Water is a crucial subsystem of the EWF nexus through which the supply of energy (e.g., hydropower), water, and food (e.g., agricultural irrigation) is attained. Therefore, the adequate assessment and forecasting of water quantities in these different sectors are of high importance [18]. The agricultural sector accounts for 3.5% to 4.8% of the total energy consumption, and 70% of the total freshwater withdrawals. It is predicted that demand for food will increase by 60% by 2050, which will lead to an increase of more than 50% of the irrigation water requirements. With this in mind, improved irrigation practices are indispensable to overcome challenges related to food security [19,20].

Critical nexus interactions are found in the supply of irrigation water requirements for agriculture. Water and energy subsystems are closely intertwined through the use of direct energy for pumping fresh water and desalinating water for irrigation. Inaccurate measurements of irrigation water requirements will thus lead to inefficient use of energy supplies [20]. Agricultural systems also consume indirect energy in the form of fertilisers, which defines a crucial energy-food nexus [21]. For example, increased food prices have been linked to spikes in fertiliser and fuel prices [19]. Thus, the type and amount of fertilisers required in agricultural systems need to be carefully assessed, monitored, and planned since assimilation of nutrients by plants varies with respect to soil moisture content and transpiration. Moreover, in cases where ET is overestimated, excess irrigation may lead to the leaching of nutrients to groundwater systems.

Nutrients and water, that have a close interaction, are directly responsible for the growth of plants and can either achieve positive or negative outcomes depending on their amounts and balance. Adequate irrigation treatments aid in nutrient availability and their transformation into useful consumable forms. The mineralisation process of organic nitrogen present in soils or from fertilisers is highly dependent on soil moisture amongst other parameters. The mineralised nitrate product from the nitrification of ammonium

increases with an increased available water content within a tolerable range as it is greatly vulnerable to leaching losses. Hence, an adequate soil moisture content is required to ensure the nitrification of ammonium, whereas an excess or a deficiency in water content restrains this process [22]. Low soil moisture can affect the amount of nutrient uptake by the plants such as sodium, potassium, calcium, magnesium, zinc, etc. This can lead to reductions at the level of the total dry weight of leaves, stems and fruits for avocado plants [23]. On the other hand, and for the same type of plant, excess soil moisture can also lead to severe repercussions on the plants associated with decreased concentrations of iron and zinc at the level of the leaves [23,24]. In accordance with these findings, citrus plants have also witnessed reductions in nutrient intake when subject to excess soil moisture content or excess irrigation regimes. Concentrations of calcium, magnesium and iron dropped in the citrus seedling leaves when soil moisture was in surplus [25]. The close interaction between nutrient efficiency and water supply makes it imperative to not only define the nutrient ratios and amounts and set their schedule but also to adequately estimate the necessary irrigation water and closely monitor its supply [22].

Calcium deficiency in plants is another consequence of low or high water availability and transpiration rates. Calcium is a nutrient that is transported via the xylem and not the phloem of the plant, which makes its movement in the different parts of the plant primarily driven by transpiration that induces a suction action to draw water and nutrients up. A high surface-to-volume ratio in fruits is an important parameter that promotes transpiration and thus helps calcium transport and its accumulation in fruits. However, as the fruit grows and becomes larger, its surface-to-volume ratio decreases and more wax deposition occurs which reduces transpiration rates. This means that as the fruit grows, it is more likely to witness reduced calcium flow into the fruit [26]. It has been proven that improving plant transpiration is more effective in solving calcium deficiencies in fruits than directly increasing calcium levels in the substrate [27]. Moreover, the effect of climate parameters, namely low solar radiation and high humidity, have been linked to calcium deficiencies in tomato plants perceived in leaf damage and reduced yields. These latter parameters drive the evapotranspiration rate which needs to be assessed in advance to predict climate variabilities and minimise their impact on crop growth and yield. In this particular study, humidity levels need to be lowered in the food system to balance the high solar intensity and enhance transpiration rates [28]. In this case, accurate estimations of transpiration will help control the flow of calcium throughout the plant and counterbalance the effect of climate stressors. Other studies revealed that excess transpiration rates were responsible for calcium deficiencies in low-transpiring species such as cauliflower, lettuce and cabbage plants. The high transpiration rates caused high calcium transport to the outer leaves at the detriment of the inner leaves to receive equitable amounts of calcium [29].

Upgrading agricultural food systems often comes at the expense of higher water and energy supply. Agricultural greenhouses are a good example of yield improvement systems, that provide a closely controlled microclimate. Temperature and humidity are some of the main microclimate parameters that can be controlled and monitored through the deployment of adequate technology systems (e.g., heating, cooling, and ventilation) which require substantial amounts of energy for their operation. These parameters are crucial drivers of plant evapotranspiration rates and thus irrigation requirements [30,31]. $CO_2$ enrichment is another yield improvement and water reduction practice in agricultural greenhouses. $CO_2$ can be procured either through purchasing commercial or industrial $CO_2$ or internally producing it using gas burners. This induces additional energy requirements and expenditures to greenhouse operations [32]. The effect of higher $CO_2$ concentrations has been witnessed in evapotranspiration reductions due to the shrinking of stomata openings in plant leaves which control gas exchange between the plant and atmosphere (i.e., water vapour and $CO_2$). Thus, it is important to account for this parameter in evapotranspiration estimates and irrigation water requirements [33,34]. Not assessing evapotranspiration based on the new microclimate settings will lead to the inefficient use of these technologies and the wasteful utilisation of energy and water resources.

## 4. Evapotranspiration Mechanistic and Empirical Models

Evapotranspiration entails the combination of two different processes; the vaporisation of liquid water from surfaces known as evaporation and that from plant tissues called transpiration and their vapour removal to the atmosphere. The evaporation process occurs at different water surfaces such as lakes, rivers, soils and vegetation and requires enough solar energy for the vaporisation of the liquid water. The vapour water removal from the water surface to the atmosphere is driven by the vapour pressure difference between the water vapour in the water surface and that in ambient air [2]. The transpiration process involves the movement of water from the soil to the plant leaves and its evaporation to the atmosphere through stomata openings within the leaves. Transpiration is an essential mechanism for plant growth and development as it enables the transportation of important nutrients and minerals through the different parts of the plant and provides the plant with the necessary cooling load for its survival. Both evaporation and transpiration processes are driven by meteorological conditions, mainly solar radiation, air temperature and humidity, and wind speed. The transpiration process is additionally influenced by crop characteristics and cultivation methods [2,15].

Potential evapotranspiration ($ET_p$) refers to the maximum amount of water that can be removed from a surface with an abounding water supply via both evaporation and transpiration. Contrary to reference evapotranspiration, potential evapotranspiration is the maximum level of evapotranspiration that can be reached from a surface under ideal conditions. Potential evapotranspiration is more applicable for analysing water demand of large areas such as water reservoirs, while reference evapotranspiration is best suited for crop assessments because it is more precise and accounts for crop-related changes. Evapotranspiration models can be classified into four main levels of description: mass transfer, temperature-based, radiation-based, and combination-based models that encompass both aerodynamics and energy balance. The mass transfer-based $ET_p$ models are founded only on aerodynamics. Temperature-based models involve only temperature as an input such as the Hargreaves and Samani model. Radiation-based models consider energy balance such as the Priestley–Taylor equation for $ET_p$ calculations. Both aerodynamics and energy balance can be aggregated in combination models such as the Penman–Monteith model for $ET_p$ and the FAO56 modified Penman–Monteith model for $ET_o$ assessments [35]. Available evapotranspiration models are either analytical where they are fully based on physical laws, mechanistic where they use physical laws to predict estimates based on causality relationships such as the original Penman–Monteith model, or empirical (statistical) where they are based on correlations developed from experimental observations such as the Hargreaves model [36–38]. Empirical models are favored for their simplicity but lack physical significance and regional accuracy [39]. When there are enough accurate available input data, the use of mechanistic approaches is more suitable than empirical models [14].

### 4.1. Penman–Monteith Equations

The original Penman–Monteith equation combines the mass transfer with the surface energy balance and is able to estimate the potential evapotranspiration rate based on meteorological data and crop physiological characteristics. This equation accounts for the surface resistance, which entails the resistance of water vapour movement through the leaf stomata and soil surface, along with the aerodynamic resistance which describes the resistance of vertical water vapour diffusion from the leaf to the surrounding air (Equation (1)) [2].

$$\lambda ET = \frac{\Delta(R_n - G) + \rho_a C_p \frac{(e_s - e_a)}{r_a}}{\Delta + \gamma\left(1 + \frac{r_s}{r_a}\right)} \tag{1}$$

where $\Delta$ is the slope of the saturation vapour pressure, $R_n$ is the net solar radiation, $G$ is the soil heat flux, $\rho_a$ is the mean air density at isobaric conditions, $C_p$ is the specific heat of air, $e_s - e_a$ represents the vapour pressure deficit, $\gamma$ is the psychometric constant, and $r_s$ and $r_a$ represent the surface resistance and aerodynamic resistance respectively.

The net solar radiation can be measured using different instruments including radiometers, pyranometers and solarimeters. In cases where net radiation data are not available, estimates can be calculated from the incoming net shortwave radiation ($R_{ns}$) and outgoing net longwave radiation ($R_{nl}$) as presented in Equations (2)–(4) [2].

$$R_n = R_{ns} - R_{nl} \qquad (2)$$

$$R_{ns} = (1 - a)R_s \qquad (3)$$

$$R_{nl} = \sigma \left( \frac{T_{max,K^4} + T_{min,K^4}}{2} \right) (0.34 - 0.14\sqrt{e_a}) \left( 1.35 \frac{R_s}{R_{so}} - 0.35 \right) \qquad (4)$$

where $R_s$ is the solar or shortwave radiation (MJ m$^{-2}$ day$^{-1}$), $a$ is the albedo coefficient, $\sigma$ is the Stephan–Boltzmann constant, $T_{max,K}{}^4$ and $T_{min,K}{}^4$ are the max and min absolute temperatures during a 24 h period, $e_a$ is the actual vapour pressure (kPa), and $R_s/R_{so}$ is the relative shortwave radiation.

The aerodynamic resistance can be expressed following the logarithmic wind profile as shown in Equation (5). Where $z_m$ represents the height at which wind speed is measured (m), $z_h$ is the height at which humidity is measured (m), $z_{om}$ is the roughness length of momentum transfer (m), $z_{oh}$ is the roughness length corresponding to heat and vapour transfer (m), d is the height (m) above the ground at which zero wind speed is attained and can be estimated as 2/3 of the obstacle's height (crop height), $k$ is the Von Kármán constant ($\sim$0.41), and $u_z$ is the wind speed at height $z$ (m s$^{-1}$). The roughness lengths included in this equation represent corrective coefficients that consider the effect of the surface roughness of the canopy on the wind profile. The roughness lengths can also be estimated as approximately one-tenth of the crop height [2].

$$r_a = \frac{\ln\left(\frac{z_m - d}{z_{om}}\right) \ln\left(\frac{z_h - d}{z_{oh}}\right)}{k^2 u_z} \; \left( \text{s m}^{-1} \right) \qquad (5)$$

The surface resistance can be estimated through a simplified equation that combines the bulk stomatal resistance corresponding to the well-illuminated leaf $r_I$ (s m$^{-1}$) and the active leaf area index LAI as shown in Equation (6). The well-illuminated leaf area corresponds to the upper part of the canopy that generally receives the most sunlight. The leaf area index LAI represents a dimensionless measure of the upper side area of the leaf per unit area of the soil underneath it. This measure depends on the plant type and density along with the growing stage. The active LAI deals with the leaf area that directly receives sunlight and contributes to the photosynthesis process, which is generally the upper part of a dense canopy. The bulk stomatal resistance represents the average resistance of a leaf and is highly dependent on the type of crop, climatic conditions and soil and irrigation water conditions [2]. It is one of the most important variables that depict the effect of crop management practices and climate conditions at the leaf level. For example, with water stress conditions such as increased water salinity, increased stomatal resistance is perceived which directly mirrors in reduced plant growth and reduced evapotranspiration rates [40]. Other models provided a more detailed estimation of this parameter, whereby variabilities in climatic conditions were integrated such as the Stanghellini model which is further discussed in the next section [41]. On the other hand, field-based measurements can provide more accurate estimates of stomatal resistances by means of measurement systems such as a leaf porometer [40].

$$r_s = \frac{r_I}{\text{LAI}_{active}} \; \left( \text{s m}^{-1} \right) \qquad (6)$$

The FAO56 Penman–Monteith is considered as the standard method for estimating reference evapotranspiration (ET$_o$) along with the crop evapotranspiration (ET$_c$) by associating a crop coefficient ($K_c$) to ET$_o$. The reference evapotranspiration model is defined

according to a hypothetical clipped grass crop of 0.12 m height, 70 s m$^{-1}$ surface resistance, and 0.23 albedo, grown under sufficient irrigation water conditions as shown in Equation (7). T represents the mean temperature of the air taken at a 2 m height. All weather data values are required to be collected at a 2 m height from the ground or need to be converted to this height for use in this model. This model is a simplification of the original Penman–Monteith equation with the introduction of assumptions specific to clipped grass as the crop under study.

$$\text{ET}_\text{o} = \frac{0.408\Delta(R_n - G) + \gamma \frac{900}{T+273} u_2 (e_s - e_a)}{\Delta + \gamma(1 + 0.34 u_2)} \tag{7}$$

$$\text{ET}_\text{c} = k_c \text{ET}_\text{o} \tag{8}$$

The crop coefficient is a representation of the physical and physiological characteristics of the crop under study relative to the reference crop, including the ground cover, canopy properties and aerodynamic resistance. The effect of these characteristics is what makes up the crop coefficient $K_c$, which can either be estimated in a single coefficient or divided into two separate effects represented by the dual-crop coefficient. The dual-crop approach consists of distinctively accounting for two different coefficients in the estimation of $\text{ET}_\text{c}$; the basal crop coefficient ($K_{cb}$) which accounts for the crop transpiration and the soil evaporation coefficient ($K_e$) which represents the evaporative losses from the soil surface (Equation (9)). The dual-crop coefficient is particularly of interest for specific applications such as when real-time irrigation scheduling is required or when high-frequency irrigation is applied [2]. The FAO56 Penman–Monteith model was developed for open-field applications. However, this model entails limitations when applied to closed agricultural mediums, such as greenhouses, due to the non-logarithmic profile of wind inside these mediums, whereby the significantly low wind speed inside (close to 0) leads to a logarithmic value of the aerodynamic resistance that tends to infinity. It is thus preferable to directly measure the aerodynamic resistance [42].

$$\text{ET}_\text{c} = (k_{cb} + k_e)\text{ET}_\text{o} \tag{9}$$

*4.2. Stanghellini Model*

The Stanghellini model revised the Penman–Monteith equation in order to have an applicable estimation of evapotranspiration for greenhouse settings (Equation (10)). This model includes the impact of the leaf area index (LAI) on the evapotranspiration rate, in which the exchange of energy from multiple layers of the canopy are considered [41,43].

$$E\lambda = \frac{\delta\, R_n + \left(\frac{2\text{LAI}\rho_a C_p}{r_e}\text{VPD}\right)}{\gamma\left(1 + \frac{\delta}{\gamma} + \frac{r_i}{r_e}\right)} \tag{10}$$

where $\lambda$ is the latent heat of vaporisation, $\delta$ is the slope of the saturation vapour pressure curve, $R_n$ is the net solar radiation, LAI is the leaf area index, $\rho_a$ and $C_p$ are, respectively, the air density and specific heat capacity, VPD is the vapour pressure deficit, $\gamma$ is the psychometric constant, and $r_i$ and $r_e$ are the canopy internal and external resistances respectively.

The net solar radiation term ($R_n$) in the Stanghellini model is described using an empirical equation that encompasses short and long wave radiation characteristics on a multi-layer canopy as shown in Equation (11). This equation also considers radiation fluxes from greenhouse surface components such as soil covering, cladding material and heating pipes [15,43].

$$R_n = 2\,\text{LAI}\left[0.07 I_s - \frac{0.16\rho_a C_p (T_h - T_0)}{r_R}\right] \tag{11}$$

where $I_s$ is the shortwave irradiance, $T_h$ is the apparent radiation temperature, $T_0$ is the leaf surface temperature, and $r_R$ is the radiation resistance. The radiation resistance ($r_R$) is expressed in Equation (12), and involves the ambient air temperature $T_a$ and the Stefan-Boltzman constant $\sigma$ [15,43].

$$r_R = \frac{\rho_a C_p}{4\sigma(T_a + 273.15)^3} \tag{12}$$

The external or aerodynamic resistance $r_e$ for the Stanghellini model is defined as shown in Equation (13).

$$r_e = \frac{\rho_a C_p l}{\lambda_a N_u} \left( \text{s m}^{-1} \right) \tag{13}$$

$$N_u = 0.37 \left[ Gr + 6.92 Re^2 \right]^{0.25} \tag{14}$$

where $l$ represents the characteristic dimension of a leaf (m), $\lambda_a$ is the thermal conductivity, $\rho_a$ and $C_p$ are the air density (kg m$^{-3}$) and air specific heat capacity (J kg$^{-1}$·C$^{-1}$). Nu is the Nusselt number and is expressed using the Grashof number ($Gr$) and the Reynolds number ($Re$) as shown in Equation (14).

As for the internal resistance, Stanghellini proposed a parametrisation equation that can properly estimate the canopy internal resistance in greenhouse settings. This equation suggests that the internal resistance of a leaf is a function of a minimum possible internal resistance ($r_{min}$) which is related to physiological aspects of the leaf in addition to the independent effect of climate parameters on the minimum internal resistance as shown in Equation (15). This method also assumes that the behavior of the leaf internal resistance is similar to that of the canopy [43].

$$r_i = r_{min}\, \widetilde{r}_i(I_s)\, \widetilde{r}_i(T_0)\, \widetilde{r}_i(CO_2)\, \widetilde{r}_i(VPD) \tag{15}$$

where $\widetilde{r}_i$ represents the relative increase in internal resistance $r_{min}$ when climate parameters such as shortwave radiation ($I_s$), leaf surface temperature ($T_0$), $CO_2$ concentration and vapour pressure deficit (VPD) vary. $\widetilde{r}_i$ functions are defined through the parametrisation Equations (16)–(20).

The effect of shortwave radiation:

$$\widetilde{r}_i(I_s) = \frac{\bar{I}_s + C_1}{\bar{I}_s + C_2} \tag{16}$$

$\bar{I}_s$ is the mean irradiance which entails the mean flux received per unit leaf area. An important observation was made by relating the irradiation to the leaf area because as the leaf area increases the available irradiation per unit area decreases which leads to an increase in leaf internal resistance. $\bar{I}_s$ is expressed by Equation (17), where $A_s$ is the coefficient of shortwave radiation [43].

$$\bar{I}_s = \frac{A_s I_s}{2\,\text{LAI}} \tag{17}$$

The effect of surface leaf temperature:

$$\widetilde{r}_i(T_0) = 1 + C_3(T_0 - T_m)^2 \tag{18}$$

where $T_m$ represents the temperature at which minimum internal resistance is achieved.

The effect of $CO_2$ concentration:

$$\widetilde{r}_i(CO_2) = 1 + C_4(CO_2 - 200)^2 \tag{19}$$

The effect of vapour pressure deficit:

$$\widetilde{r}_i(VPD) = 1 + C_5(VPD)^2 \tag{20}$$

Parameters $C_1$–$C_5$ were determined by an optimisation model that produced the best-fit combination of parameters for the observed internal canopy resistance for three consecutive days as summarised in Table 1.

**Table 1.** Internal resistance parameters for the Stanghellini model.

| Coefficient | Daytime Value | Nighttime Value |
|---|---|---|
| $C_1$ (W m$^{-2}$) | 4.30 | - |
| $C_2$ (W m$^{-2}$) | 0.54 | - |
| $C_3$ (K$^{-2}$) | $2.3 \times 10^{-2}$ | $0.5 \times 10^{-2}$ |
| $C_4$ (vpm$^{-2}$) | $6.1 \times 10^{-7}$ | $1.1 \times 10^{-11}$ |
| $C_5$ (kPa$^{-2}$) | 4.3 | 5.2 |
| $T_m$ (°C) | 24.5 | 33.6 |
| $r_{min}$ (s m$^{-1}$) | 82 | 658.5 |

Similar to the Stanghellini model, [44] derived an evapotranspiration model for greenhouse settings that incorporated a canopy area index (CAI) as a simplification of the irradiance term instead of the net solar radiation empirical calculation proposed by Stanghellini.

### 4.3. Priestley-Taylor Model

The Priestley–Taylor model is useful when parameters defining the aerodynamic resistance are unavailable. This method offers a dimensionless coefficient $\alpha$ that can replace the aerodynamic resistance term in the Penman–Monteith equation (Equation (21)). Priestley and Taylor suggested that an average value of 1.26 for $\alpha$ is fairly reasonable [45,46]. Another study developed an adaptation equation for $\alpha$ based on the daily mean vapour pressure deficit as presented in Equation (22) [12].

$$\text{ET}_\text{P} = \frac{1}{\lambda} \Delta \frac{R_n - G}{\Delta + \gamma} \alpha \tag{21}$$

$$\alpha' = 1 + (\alpha - 1)\text{VPD} \tag{22}$$

However, limitations of the original Priestley–Taylor method under advective conditions are perceived as underestimates of evapotranspiration rates [14]. Advective conditions occur when atmospheric properties of air such as vapour and heat are transported to the crops by wind movement. Hence, the elimination of the aerodynamic resistance parameter restricts the spatial applicability of this model [47]. Moreover, studies uncovered interactions between the $\alpha$ coefficient and other climatological and physiological parameters such as soil moisture content, solar radiation, atmospheric stability, etc. For example, as the surface resistance or aridity of the region increases, the coefficient $\alpha$ increases [48,49]. Other studies intended to calibrate the Priestly–Taylor model on the basis of the PM model. [48] found that an $\alpha$ value of 1.26 is low for the region under study and suggested a value of 1.82 for cold climates and 2.14 for arid climates to achieve better ET estimates. Moreover, Priestley–Taylor modified versions have been suggested in the literature, whereby the equations accounting for a crop coefficient and varying alpha coefficients based on surface temperature were developed and included in the initial model [50].

### 4.4. Hargreaves and Samani Model

The lack of available meteorological data and issues related to their quality and accuracy, especially in developing countries, can pose limitations in the use of certain ET models such as the Penman–Monteith [48]. The Hargreaves and Samani model represents an equation through which the global solar radiation at the surface, *Rs*, is estimated through air temperature values (Equations (23) and (24)). This equation can be used to estimate ET values when net solar radiation data is unavailable or questionable in terms of accuracy [2,51,52].

$$R_s = k_{RS}\sqrt{T_{max} - T_{min}}\,R_a \tag{23}$$

$$ET_0 = 0.0023 \left( \frac{T_{max} + T_{min}}{2} + 17.8 \right) \sqrt{T_{max} - T_{min}} R_a \tag{24}$$

where $R_a$ is the extra-terrestrial solar radiation, $T_{max}$ and $T_{min}$ are the maximum and minimum air temperatures respectively, and $K_{RS}$ is an empirical adjustment coefficient which depends on the site location. $K_{RS}$ takes a value of 0.16 °C$^{-0.5}$ for interior regions and 0.19 °C$^{-0.5}$ for coastal regions. $K_{RS}$ is fitted to $R_s/R_a$ versus ($T_{max} - T_{min}$), and usually increases with increasing temperature.

These constants limit the model to specific sites and can engender overestimations of ET rates which in turn can lead to excess irrigation. Hence, several studies investigated the validity of the Hargreaves model under various locations and suggested calibration parameters, which helped in reducing the overestimation of ET values [53]. For example, a calibration conducted by [54] decreased the overestimation of ET by 16.3%.

## 5. Evapotranspiration Measurement Techniques

### 5.1. Leaf Area Measurements

Leaf area index measurement techniques are divided into two main categories: direct and indirect. The direct measurement systems entail destructive approaches from harvesting leaves. Indirect leaf area measurement systems are non-destructive techniques through which the leaf area is estimated by assessing how the canopy intercepts radiation [55]. Leaf area estimates are important because they reflect the transpiring surface size. This estimate can be integrated into the models discussed previously to depict variations in the internal resistance and net radiation within the multi-layered canopy. Ceptometers are a cost-effective tool that draws an estimate of the leaf area index (LAI) by measuring the photosynthetically active radiation (PAR) above and below the canopy. Several studies investigated the accuracy of the ceptometry technique against destructive methods and concluded that it provides good accuracy for LAI measurements of uniform canopies [56]. Another tool for indirect LAI measurement is via hemispherical photography. This technique involves the study of canopies via fisheye shaped lenses located downward (looking up) or upward (looking down) the canopy. It provides information about the size, density, position and distribution of gaps detected in the canopy. However, this technique necessitates extensive post-processing of each image independently which can lead to errors [17,57]. The leaf area meter such as the LAI-2200C proposed by Licor is another technique, which measures the interception of blue light from below and above the canopy [57]. Image-based remote sensing techniques can also be considered as an indirect measurement of LAI, which are based on estimating LAI from empirical relations between LAI and vegetative indices [58].

### 5.2. Leaf Temperature Measurements

Estimating surface leaf temperature can enhance model-based estimates of ET rates. Leaf temperature and the temperature gradient between the leaf surface and the ambient directly impacts the rate of transpiration. Under ideal conditions, the temperature at the leaf surface is lower than that of the ambient. The opposite, either higher leaf temperature or equal to the ambient, is an indication of crop stress and unsuitable growing conditions. Hence, it is crucial to have an estimate of surface leaf temperature as it defines the saturation water vapour concentration within the stomata which represents an indication of gas exchange between the leaf and the atmosphere [59]. Thermocouples are thermoelectric systems based on converting a temperature signal into an electric signal. The main advantages of this system are its low cost, simple operation, light weight, and fast response as compared to other more complex measurement techniques [60]. However, the main disadvantage of this system is the direct contact with the leaf surface, in which the thermocouples can absorb solar radiation and heat from the leaf by conduction. These problems lead to significant errors in leaf temperature estimates [61]. Infrared thermometers are also used for leaf temperature measurements and consist of infrared temperature sensors that measure the infrared energy emitted by a specific spot on the leaf surface and transform

it into a measurable electrical signal. The major advantages of this method, apart from being contactless, are quick response and high accuracy. However, the infrared method is sensitive to the environment in which dust and steam can significantly influence its precision [60]. Thermal infrared imaging is another leaf temperature measuring system that contains an optical system and is considered a remote sensing technology. This system entails temperature measurements at multiple points on the leaf surface, as opposed to spot infrared thermometers. The infrared camera uses infrared detectors that are sensitive to wavelengths between 7–14 μm to capture infrared energy and converts it into two-dimensional thermographic image visualisations. These cameras are able to evaluate temperature gradients over large temporal and spatial scales contrary to thermocouples, which can help identify variations in ET across large crop areas. Impacts of some parameters such as the leaf surface emissivity and the longwave radiation can engender some inaccuracies in temperature measurements. Although most integrated software accounts for a correction factor for these variables, it is mostly based on indoor controlled conditions such as in laboratory settings. Some software can also have fixed corrective factors which cannot be changed by the users depending on their outdoor conditions. Other software can combine user-inputted corrective parametrisations, however, it is usually challenging to estimate these factors due to the complex settings and dynamic environmental variables [62]. Moreover, thermal cameras hold high acquisition costs, which limit their use in agricultural applications [60].

### 5.3. Eddy Covariance Systems

The eddy covariance is considered as one of the techniques for the direct measurement of evapotranspiration. The eddy covariance is comprised of two sections: an anemometer that directly measures wind speed and direction, and an infrared gas analyser (IRGA) that measures gas concentrations in the air such as water vapour. The simultaneous evaluation of changes in vertical air velocity and water vapour concentration in the air is what enables the measurement of evapotranspiration in the form of a vertical flux of water vapour. The eddy covariance technique has been applied to open field applications including field crops, forests, water bodies, and grasslands, etc. [63]. The eddy covariance method entails challenging operations as it involves high-frequency measurements along with complex processing of simultaneously collected data. Moreover, the validation of ET estimated by this system with other methods is quite challenging due to the large scale, highly variable area and open boundary layer of the volume studied which does not achieve energy balances [64].

### 5.4. Weighing Lysimeters

Weighing lysimeters directly measure evapotranspiration by evaluating changes in the mass of the soil and crop. They necessitate that the soil structure and composition, the physiological characteristics of vegetation (e.g., height), and the climatic conditions of the growing medium inside the lysimeter are similar to the ones outside the lysimeter. The high economic cost and intensive installation and maintenance requirements of lysimeter systems limit its application on various parts of the agricultural system to perceive spatial evapotranspiration variations, which restricts measurements to be taken on only one or few parts of the land under study. However, a significant advantage of lysimeters is that they provide simultaneous information on percolation of excess irrigation and soil-water retention that no other methods provide [64].

### 5.5. Gas Exchange Measurement Systems

Gas exchange measurement systems enable direct and accurate estimates of ET rates by tracing the absorption of gases through an infrared light source (infrared gas analyser IRGA). The latest advancements of these systems operate under open chambers that evaluate differential gas exchanges through estimating the difference in gas concentrations (i.e., $H_2O$ and $CO_2$) between the input and output of the chamber [65]. These systems

can also estimate $CO_2$ exchanges at the stomata level, which can be crucial in estimating the impact of increased $CO_2$ concentrations in the air on the internal resistance and on the evapotranspiration levels [66]. Higher $CO_2$ concentrations can be caused by increased greenhouse gas emissions or linked to $CO_2$ enrichment practices in greenhouse settings. Particularly in the latter application, evaluating the effect of varying $CO_2$ concentrations is of high importance to determine the ideal $CO_2$ concentration that needs to be injected for optimal outputs in terms of yield and evapotranspiration levels [67].

*5.6. Remote Sensing*

Remote sensing revolves around the observation and measurement of parameters without physical contact with the subject under study. Remote sensing data can be retrieved from satellite technologies and provide information about biophysical parameters that can assess evapotranspiration such as the type and density of the vegetation and the surface albedo [16,68]. Several approaches have been developed in the literature for estimating evapotranspiration from remote sensing data, from which two general methods have been widely used in the agricultural field. One method uses radiometric surface temperature to separate latent heat from sensible heat. The second approach is based on vegetation indices (VI), taken from surface reflectance, that can estimate basal crop coefficients on spatial scales. Vegetation indices from satellite remote sensing include leaf area index (LAI) and the normalised difference vegetation index (NDVI) which can be included within the surface resistance estimation in the Penman–Monteith model [69,70]. The radiometric temperature can be adjusted to determine aerodynamic temperatures through semi-empirical or empirical models that incorporate spatial distribution in surface roughness lengths. This can be included in the Penman–Monteith equation to estimate ET rates. As for the basal crop coefficient, it can be used to estimate crop evapotranspiration from the reference evapotranspiration [69,71]. ET estimates from remote sensing data offer a large spatiotemporal distribution, which makes it a prevailing method in large scale and climate impact mapping applications [72]. However, challenges remain to obtain reliable estimates in regions with cloud cover and dust. Various studies tackled the reconstruction of missing data in these regions by means of different methods such as cloud removal and gap filling, but the linearity of these models still poses some limitations [73]. Remote sensing data are also used in data-driven models which estimate evapotranspiration rates by different data forcing methods such as machine learning, regression, neural networks, etc. The data-driven models can also be coupled with physical models to parametrise certain subprocesses dealing with uncertainty [74].

## 6. General Directions for Evapotranspiration Estimates

Evapotranspiration models differ in terms of application, inputs needed and time-step as summarised in Table 2. The mechanistic and empirical models can also be coupled with direct and indirect measurements depending on the application and available measurement systems as illustrated in Figure 3. The original Penman–Monteith equation is considered a powerful model for the estimation of evapotranspiration as it combines both the aerodynamic and surface resistances. However, limitations of this model occur in the calculation of the surface resistance which entails intensive data collection and modeling. With the FAO56 simplified Penman–Monteith model, the surface resistance term is replaced by a fixed term following a reference crop and standard conditions. However, the simplification of the surface resistance term can reduce the accuracy of ET estimates. The Priestley–Taylor model is a good model for potential evapotranspiration calculation when aerodynamic data are unavailable, hence requiring fewer input data. However, underestimates of ET rates have been reported using this model under advective conditions. Similarly, the Hargreaves and Samani model requires less input data and is primarily beneficial when meteorological data are unavailable especially net solar radiation. However, the integration of an empirical constant in this model can lead to overestimations of ET rates and weaken the accuracy of this model.

**Table 2.** Summary of evapotranspiration models (adapted from [46]).

| Evapotranspiration Model | Application | Reference Crop | Time Step | Inputs |
|---|---|---|---|---|
| Penman–Monteith | Open field | Clipped grass and Alfalfa | Daily and hourly | Solar radiation, air temperature, relative humidity, and wind speed. |
| Stanghellini | Greenhouse | Tomato | Hourly | Solar radiation, air temperature, relative humidity. |
| Priestley–Taylor | Open field | - | Daily | Air temperature and solar radiation. |
| Hargreaves and Samani | Open field | - | Daily | Air temperature. |

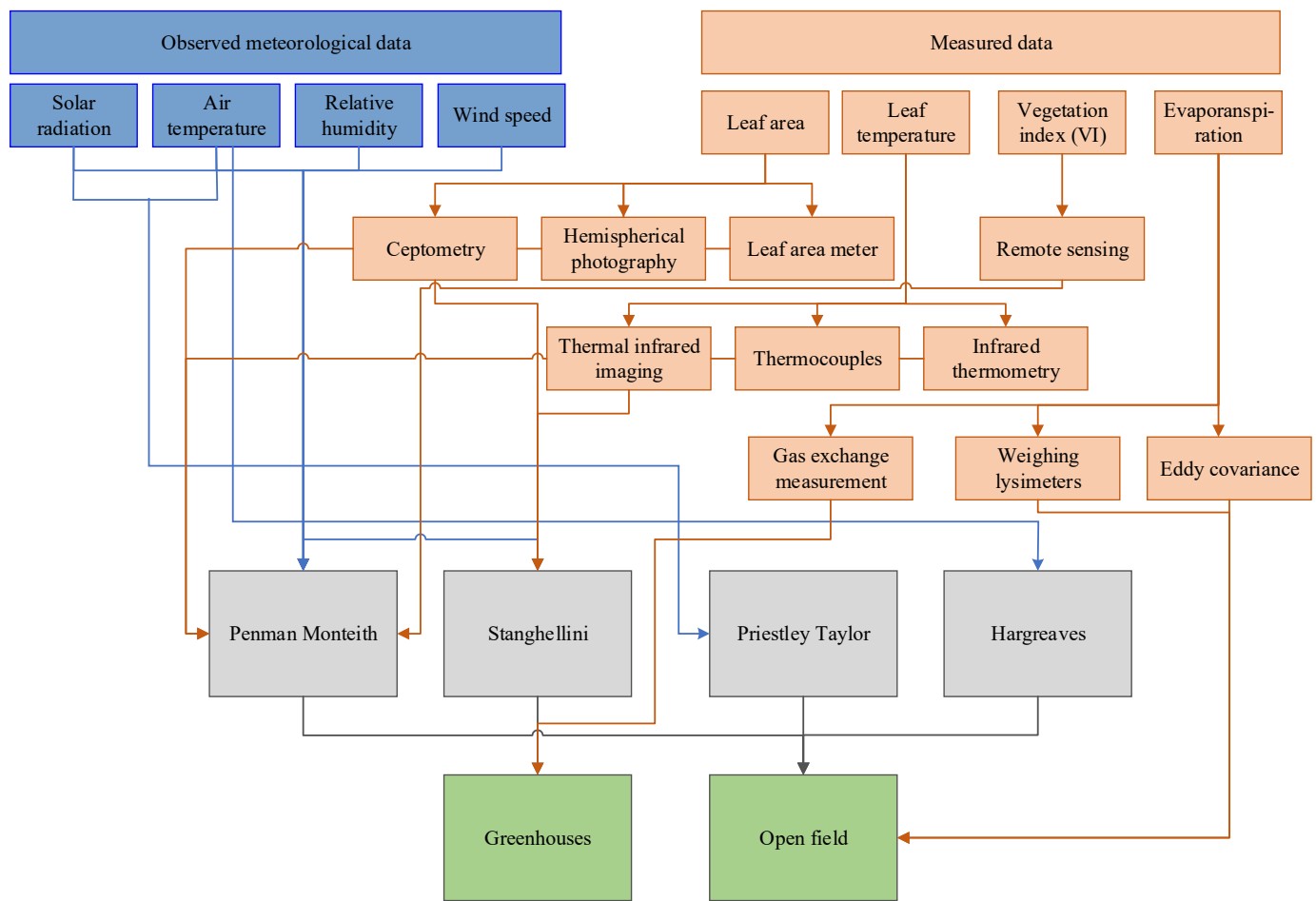

**Figure 3.** Directions for evapotranspiration measurements.

Several studies compared existing mechanistic and empirical models, especially for greenhouse applications as summarised in Table 3. [15] compared ET estimates from the Penman–Monteith and Stanghellini models and that of direct ET measurements for greenhouse settings. Findings of this study showed an enhanced ET prediction with the Stanghellini model with a model efficiency ($R^2$) of 0.872 as compared to 0.481 achieved through the Penman–Monteith model [15]. Overestimations of the evapotranspiration rates were observed, and few studies explained the potential cause of these overestimations including the challenging parametrisation of the canopy and aerodynamic resistances and the somewhat non-homogeneous microclimate data collected in greenhouse environments [13]. The Penman–Monteith model assumes homogeneity in collected climatic data which can

be appropriate for open field settings but can rather introduce discrepancies for ET estimates in greenhouse mediums. [75] investigated the use of the Penman–Monteith model in greenhouses by comparing its ET estimate to direct measurements. The findings of this study demonstrated that the location of climatic parameters collected has a large impact on ET estimates. This study further suggests the adaptation of the Penman–Monteith model to greenhouse conditions, whereby if temperature and humidity measurements are taken inside the canopy rather than on top of the canopy, the ET model shows improved accuracy for greenhouse applications [75]. This method can be applied by adopting accurate measurement systems for the estimation of leaf surface temperatures such as thermocouples, infrared thermometry and infrared thermal imaging.

On another note, the consideration of the leaf area index factor and the net solar radiation empirical equation in the Stanghellini model explain the improvements in evapotranspiration estimates for greenhouse environments, where the multi-layered leaf aspect of the plant canopy is considered. Moreover, the inclusion of the effects of microclimate variations in the internal canopy resistance provides a better estimate for ET rates in greenhouses. This consideration offers many benefits that can also be applied to open field agricultural systems, with the appropriate parametrisation, whereby effects of climate variations can be examined on the stomatal level and included in the evaluation of the canopy internal resistance. Detailed studies have also successfully proven that the Stanghellini model gives accurate estimates for unheated greenhouses with natural ventilation [76], and for cooled greenhouses with natural ventilation and high pressure fogging [41].

**Table 3.** Studies comparing evapotranspiration models.

| Study | Aim | Conclusion |
|---|---|---|
| [15] | Comparison between four evapotranspiration models against direct ET measurements for greenhouse settings: Penman, Penman–Monteith, Stanghillini, and Fyn models. | The Stanghellini model has the best model performance for ET predictions. |
| [77] | Comparison between Penman–Monteith and Stanghellini for greenhouse grown tomato crops. | The Stanghellini model has a better estimate due to the LAI, net radiation and stomatal resistance considerations. |
| [41] | Comparison of three evapotranspiration models for two crops grown in greenhouses with cooling. | Overestimation of ET by the Penman–Monteith model.Need for parameter adjustments for Penman–Monteith and Stanghellini models. |
| [3] | Comparison of six evapotranspiration models in an open field agricultural system. | Direct methods such as the original Penman–Monteith model propose better estimates than indirect methods such as the FAO56 model. |
| [78] | Comparison of the Penman–Monteith, Priestley–Taylor and Hargreaves models for a specific location. | The Priestley–Taylor and Hargreaves models underestimate ET values. |

The Penman–Monteith model is recognised as the basis for all ET mechanistic and empirical models. The FAO56 simplified Penman–Monteith was developed as a standard for irrigation management and can be considered as a general method for ET estimates under open field conditions. On the other hand, if accurate ET estimates depicting the effect of climate variations (such as increased ambient $CO_2$ concentrations) or growing conditions (such as salt stress) on the leaf level, the original Penman–Monteith is advisable. Instead of adopting the FAO56 assumptions related to internal and aerodynamic resistances, direct measurements of the stomatal and boundary layer resistances can be

conducted and integrated into the original Penman–Monteith equation [40,79]. These measurements can also be used to calibrate existing models to specific climatological and growing practices [13]. Measurement systems include weighing lysimeters, eddy covariance and gas exchange systems that can estimate ET rates, however, these systems are associated with high capital and maintenance costs. Moreover, other measurements such as surface leaf temperature and leaf area index are recommended by means of different systems [36]. However, the economic cost related to the acquisition and maintenance of direct measurement systems can curtail their use by profit-based agricultural organisations. These systems are mainly used for research purposes to develop parametrisation improvements into existing ET models based on the location and its meteorological conditions, the type of crop grown, and growing conditions and practices [3]. Finally, the Stanghellini model is the best-adapted model for greenhouse settings. It is also advisable to adapt this model to the greenhouse's architecture (e.g., screen material, land covering, roof shape), specific growing conditions (e.g., ventilation, cooling, heating), type of irrigation (e.g., hydroponics), crop type, etc. Hence, there exists a trade-off between the high economic cost of acquiring direct measurement systems and improving the accuracy of ET estimates.

## 7. Conclusions

This review presents five major evapotranspiration (ET) models, describes their underlined assumptions and evaluates their applicability for open and closed agricultural systems. This study also highlights the importance of parameter measurements that enter in the estimation of the evapotranspiration such as leaf area index (LAI) and surface leaf temperature, in addition to direct and indirect evapotranspiration measurements. Directions for estimating evapotranspiration rates are proposed based on the reviewed models and measurement techniques for both open and closed agricultural settings. In general, the FAO56 Penman–Monteith equation presents a simplified and fairly accurate estimate for open field ET estimations. The Priestley–Taylor and Hargreaves models are a suitable approach where the extensive data requirements of other models are unavailable. The Stanghellini model best describes the evapotranspiration flux exchanges in greenhouses, whereby the multi-layered canopy aspect is considered along with a microclimate-based internal resistance estimate, and an LAI-based net solar radiation calculation. Adaptation of these models to climate variations and growing practices (e.g., irrigation type and scheduling) is necessary for improved accuracy and optimal supply of irrigation requirements. Additionally, adequate evapotranspiration measurements provide nexus opportunities that alleviate major trade-offs within the energy, water and food sectors for agricultural applications.

**Author Contributions:** Conceptualization, I.G., H.R.M. and T.A.-A.; methodology, I.G.; writing—original draft preparation, I.G.; writing—review and editing, I.G., H.R.M. and T.A.-A.; project administration, T.A.-A.; funding acquisition, T.A.-A. All authors have read and agreed to the published version of the manuscript.

**Funding:** This research is supported by the Qatar National Research Fund proposal (NPRP11S-0107-180216).

**Institutional Review Board Statement:** Not applicable.

**Informed Consent Statement:** Not applicable.

**Conflicts of Interest:** The authors declare no conflict of interest.

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
