# Peer review of "A Review of Evapotranspiration Measurement Models, Techniques and Methods for Open and Closed Agricultural Field Applications"

_water, doi:10.3390/w13182523_

Round 1

Reviewer 1 Report

Manuscript Number: water-1311076 Title: A review of evapotranspiration measurement methods, models and techniques for open and closed agricultural field applications The authors reviewed the major analytical and empirical models for estimating evapotranspiration including the Penman-Monteith, Stanghellini, Priestly Taylor and Hargreaves models. However, there are some concepts that are not clear. Therefore, I recommend that this paper can be published after major revisions indicated below. Comments - The author mentioned the evapotranspiration model in the paper, but did not specify whether it is a potential or reference crop evapotranspiration model. Although both potential and reference crop evapotranspiration provided estimates of atmospheric evaporative demand, they are based on different concepts and have different equations. Therefore, the author should be clear about this. (Xiang K, Li Y, Horton R, et al. Similarity and difference of potential evapotranspiration and reference crop evapotranspiration–a review[J]. Agricultural Water Management, 2020, 232: 106043) - There were many evapotranspiration models, why did the author choose the Penman-Monteith, FAO56 Pen-89 man-Monteith, Stanghellini, Priestley-Taylor and Hargreaves models? - The three levels of description can be placed inside the triangle in the Figure 1. - Although the author introduced five evapotranspiration models, their applicable areas and application conditions were not summarized. The advantages and limitations of each model are not summarized. - The author should add a table, which was used to summary the advantages and limitations of evapotranspiration measurement techniques for direct measurements.

Reviewer 2 Report

 A review of evapotranspiration measurement methods, models and techniques for open and closed agricultural field applications

There are some points described below that have to be considered before publication. For instance, though authors have mentioned the literature survey part, it fails to provide clear view on the previous attempts that have used Hargreaves based  ET0 estimates very similar to those from the FAO-56 PM approach. There are some points described below that have to be considered before publication. The overall presentation in the Introduction section lacks synergy and exists in bits and pieces. Though authors have identified the research gaps the literature survey part can be more streamlined and while coming towards the problem statement. The introduction section need some rework and restructuring.

-There is a need to clearly state the objective(s) of the study towards the end of the introduction.

- Discussion and Conclusion sections can be more rigorous with objective base.

I have a big concern in the Introduction, as the authors have missed providing detailed discussion on the important aspect of different classification of ET estimation methods. There is a vast literature on this I would like to suggest few lines following this which author should add is “The ETo estimation models available in the literature may be broadly classified as (1) fully physically-based combination models that account for mass and energy conservation principles; (2) semi-physically based models that deal with either mass or energy conservation; and (3) black-box models based on artificial neural networks, empirical relationships, and fuzzy and genetic algorithms”. I would recommend adding these recent references to add more scientific weight in their Introduction.

Srivastava, A., Sahoo, B., Raghuwanshi, N. S., & Singh, R. (2017). Evaluation of variable-infiltration capacity model and MODIS-terra satellite-derived grid-scale evapotranspiration estimates in a River Basin with Tropical Monsoon-Type climatology. Journal of Irrigation and Drainage Engineering, 143(8), 04017028. https://doi.org/10.1061/(ASCE)IR.1943-4774.0001199

Srivastava, A., Sahoo, B., Raghuwanshi, N. S., & Chatterjee, C. (2018). Modelling the dynamics of evapotranspiration using Variable Infiltration Capacity model and regionally calibrated Hargreaves approach. Irrigation Science, 36(4), 289-300.

Potential evapotranspiration and reference crop evapotranspiration differ in their developments, concepts, equations and application fields, however, authors have mixed the utilization of the two terms. Thus, it is necessary to clarify the terms to guide their proper usage. The aim of this study is not clear which then is needed to provide a comprehensive review of the concepts, developments, equations and applications. The review does not shows clearly the concepts and developments of evapotranspiration methods by using several other techniques which have been mentioned in previous researches.

This review has several flaws that does not serves to clarify the origins, definitions, and uses of ET. It does not addresses common ambiguities between the several other recently utilised methods. There are several other method which are used to evaluate and estimate the crop coefficient authors have not provided any sort of recent references to gain the interest of the readers.

There are some points described below that have to be considered before publication. For instance, authors have missed discussing the ET reviews by using MODIS product to evaluate the ET, there are several studies that have stated the biases which authors have missed therefore it would be great to include those studies.  I would highly refer and recommend to add the recent reference given here (https://doi.org/10.1007/s11269-020-02630-4 in Water Resource Management) from which authors can benefit.

Reviewer 3 Report

I really found reading this manuscript interesting as authors have done a thorough review of evapotranspiration measurement methods, models and techniques for open and closed agricultural field. This research is well conducted, however there are few things that I would like to recommend authors, in order to improve the quality of the manuscript before considering it for publication. Therefore, I am giving minor revision and thereafter addressing those, it can be considered for publication.

Authors need to add overall contribution form this review article in the end of abstract.

Paragraphs are not uniformly distributed I suggest to merge few paragraphs for better organization. 

Extensive English editing is required as there many problems with sentence restructuring, grammatical errors, punctuations. I suggest authors to consider the English editing a serious concern in this manuscript and with the help of native speaker they can improve this version of the manuscript adequately.

In Figure 2, authors need to explain about different colour they have used, is it random or they have some message or category to it.

Further, authors can discuss some of the studies that have corrected Hargreaves with FAO and evaluated the FAO with missing data and add the following citations: 

Kumari, N., Srivastava, A. (2020). An Approach for Estimation of Evapotranspiration by Standardizing Parsimonious Method. Agric Res 9, 301–309.

Elbeltagi, A., Kumari, N., Dharpure, J. K., Mokhtar, A., Alsafadi, K., Kumar, M., ... & Kuriqi, A. (2021). Prediction of combined terrestrial evapotranspiration index (CTEI) over large river basin based on machine learning approaches. Water, 13(4), 547.

I highly recommend authors to write the key message from the conclusion in bullet points.

Round 2

Reviewer 1 Report

(1) Title: evapotranspiration is a big word. The manuscript reviewed advances of different kinds of evapotranspiration. However, evapotration includes actual, crop reference, pan, potential or others. the title should make clear which kind of evapotranspiration you referr to. (2) From the title to the inside of the manuscript, you shold strictly differ different kinds of vapotranspiration. Because a review paper should lead the readers to correct understanding of the concepts, the equations, etc. (3)PM , Stanghellini, Priestly Taylor equations are for potential evapotranspiration. Hargreaves and Samani is for crop reference evapotranspiration. Please clarify in the manuscript. (4) Organization. In the title "A review of evapotranspiration measurement methods, models and techniques for open and closed agricultural field applications", you first mention measurements, then models, then techniques. So your organizastion should be :1. introduction, 2 measurement methods, 3. models, 4, techniques, 5, conclusion. The current structure is not good. (5) A review paper should comprehensively include a lot of international papers or books or reports. In addition, you should also include papers in 2021 or new references. I suggest you read more papers and make the paper more informative. (6) There are only 4 models mentioned in the mansucript. THere are a lot of references related to models and revised models. Xiang et al. (2020) have reviewed and differed the ETp and ETo, please refer to this mansucript and show something different with them. I suggest the authors concentrated to PM and revised PM (there should be many equations that were revised version of PM in the world, not just the one you mentioned , namely Stanghellini in the mansucript) .

Reviewer 2 Report

Dear Authors,

I want to thank the authors for addressing previous comments and for their constructive work. I found all replies satisfactory and the changes made to the manuscript significantly improve the quality of the paper. The majority of the comments are addressed and the review of the evapotranspiration is well updated and structured with adding important works. Overall, I highly recommend the resubmitted manuscript for publication in the prestigious Water journal.

Author Response

We would like to thank the reviewer for their comments which have tremendously helped enhance this manuscript.